# Should I Reuse It or Throw It Out? Analysis of the Management of Household Plastic Waste by Brazilian Consumers during the COVID-19 Pandemic through Practice Lens

**Aline Ribeiro Gomes** [1] **, Jose Carlos Lazaro** [2,*] **and Aurio Leocadio** [2]

[1] Department of Administration, School of Economics Business and Accounting, Faculty of Economics, Administration and Accounting, Butantã Campus, University of São Paulo, São Paulo 05508-010, Brazil; alineribeiro@usp.br

[2] Postgraduation Program on Management and Controlling—PPAC, Faculty of Economics, Administration and Accounting, Campus Benfica, Federal University of Ceará—UFC, Av. Da Universiade 2431, Benfica 60180-020, Brazil; aurio@ufc.br

\* Correspondence: lazaro.ufc@gmail.com or lazaro@ufc.br; Tel.: +55-85-96186229

**Abstract:** This study aims to examine the impact of the COVID-19 pandemic on the reuse of plastic bags from supermarkets among Brazilian consumers through the lens of practices. This qualitative research took place through the collection of records in digital 'solicited diaries' with interviews, using autoethnographic field diaries. The analysis process of the collected data took place through content analysis as proposed by Bardin (2014). From the results obtained, there was a greater tendency to dispose of plastic supermarket packaging and different performances to maintain this practice. These new procedures involve an increase in the consumption of cleaning products, such as bleach and soap, the addition of alcohol in this disinfection routine, and an increase in water consumption, which signals major impacts on the environment through the use of a natural resource in danger of scarcity and the release of polluting substances. The changes underwent in the performances invariably culminate in environmental impacts, either on the disposal of the plastic bags or in their hygiene for later reuse. These results alert to the challenges that governments, institutions, and individuals will have to face in an attempt to reverse the damaging effects of the pandemic on sustainability goals. Also, it contributes to consumer behavior in crisis-context literature, just as concerning the waste household.

**Keywords:** plastic waste; COVID-19; sustainability; practice theories; consumer behavior

## 1. Introduction

The spread of the SARS-CoV-2 virus in the world has given rise to a global pandemic with profound social, economic, and, above all, health impacts [1]. The restrictions resulting from combating the dissemination of COVID-19 have been challenging on different issues, including the management of Urban Solid Waste (USW) [2], with considerable consequences, mostly negative, in the practices of reuse and reduction [3].

In Brazil, in a period before the pandemic, according to the Brazilian Association of Public Cleaning Companies and Special Waste (Abrelpe, i.e., Associação Brasileira de Empresas de Limpeza Pública e Resíduos Especiais) [4], 380 kg of waste was generated by each individual in 2018, resulting in 79 million tons. Only 59.5% (43.3 million tons) of the final amount of waste collected in the same year was disposed of in landfills [4] and only 67,048 tons, on average, were collected by cooperatives and associations for processing of recycling [5].

In forecasts prepared in a scenario before the pandemic, the generation of waste already showed an increasing trend for the following years [4], since population growth and increased per capita consumption in urban areas and developing countries tend to boost waste generation [6]. With this new worldwide phenomenon, projections of a considerable increase in the quantity of household waste are already emerging [7].

Plastics play a leading role in the household waste generated: from its emergence, around 1950, until 2015, it is estimated that 6300 million tons of plastic waste were generated worldwide. A total of 79% of this waste was accumulated in landfills or the natural environment [8]. The bags, considered single-use plastic items, have a shelf life ranging from less than one year to less than three years [9] and it is estimated that between 500 billion and one trillion bags are consumed annually worldwide [10].

Given the increase in home delivery services for products considered essential in a pandemic context, there has been an increase in the consumption of disposable plastic bags [11]. Generally, plastic supermarket bags are composed of high-density polyethylene and organic and inorganic additives with a great impact on the environment [12]. However, their characteristics of resistance, low cost, and hygienic form freight transport [13] make it difficult to replace them with other short-term alternatives [14].

The reuse of plastic supermarket bags presents itself as one of the alternatives to minimize their impact. There is an impossibility of their total replacement in a short period, since this requires more complex alternatives aimed at a critical evaluation of their entire cycle of life, touching on aspects such as waste prevention initiatives, efficient resource management, reduction of the environmental footprint, and political decision-making [14].

Plastic bags symbolize a commitment to sustainable consumption in which behavioral changes in this regard can represent a major contribution to the achievement of a more sustainable future [15], Given that the focus on human behavior is essential to search for solutions based on a multidisciplinary perspective, such as the plastics problem [16]. Thus, the study of consumption habits changes offers opportunities to check the possibilities of aligning their practices inherent to their production and consumption patterns to achieve sustainable development.

Practices related to sustainable consumption are addressed in the Practice Theories in works such as those by Spaargaren [17], Hargreaves [18], and Sahakian e Wilhite [19]. However, in a pandemic context, this theory is still in an embryonic stage. Considering this circumstance and its significance for portraying a phenomenon in real-time, this research applies the lens of the practices to answer the following problem: how has the COVID-19 pandemic affected the reuse of plastic supermarket bags? Thus, its aims to examine the impact of the COVID-19 pandemic on the consumption and reuse of plastic bags from supermarkets among Brazilian consumers through the lens of practices.

To this end, an ethnographic-inspired survey was carried out based on the experiences of 30 Brazilian informants reporting their changes in bundle of practices of consuming and reusing plastic bags from supermarkets due the new situation brought by the COVID-19 pandemic.

This article provides an overview of the process of changing consumer behavior concerning the treatment given in the reuse of plastic bags in the situation of quarantine, isolation, and social distance resulting from the pandemic of COVID-19, thus contributing to the literature of consumer behavior in a crisis context, as well as concerning household waste.

## 2. Theoretical Background

### 2.1. Consumption of Plastic Grocery Bags

Dunn, Caplan, and Bosworth [20] point out that plastic supermarket bags are present everywhere. Among other types of disposable plastic products, they are produced for single handling before disposal or recycling [9]. Plastic supermarket bags generate high amounts of waste, in addition to causing long-term problems in the environment when their disposal is performed improperly [21]. However, in some homes, this kind of bag is reused, and this reuse constitutes a way of reducing their environmental [14], social, and economic [15] impact.

Reuse is one of the actions that is part of the five Rs policy, which aims to contribute to minimizing the damage generated by waste from consumption through the awareness of individuals and is composed of the following actions [22]: (a) rethinking need for consumption and production and disposal standards adopted; (b) refuse possibilities

for unnecessary consumption and products that generate significant impacts; (c) reduce to avoid waste, consume fewer products, giving priority to those that offer less waste generation potential and have longer durability; (d) reuse so that everything that is in good condition is reused; (e) recycling to provide the transformation of used materials into raw materials for other products.

In a COVID-19 pandemic context, there is the possibility of increasing the use of plastic bags. It is estimated that the production rates of household waste rose from 15 to 25% compared to the previous year due to quarantine, isolation, and social distance measures [7]. Plastic supermarket bags are included in these residues. The practices of their reuse also face the challenge of being compromised, since from this situation the processes inherent to consumption are subject to substantial disturbances [23]. This unprecedented condition can lead to increased spending on items [24], a large generation of household waste [25], and increased disposal of plastic packaging [26].

In addition to Patrício Silva et al. [26], which investigates the need to readjust plastic waste management policies in this period, this concern of the volume of plastic waste generated as a result of the COVID-19 pandemic is addressed by authors such as Klemeš et al. [2], who focus on the possibilities that the pandemic can point to as a catalyst in the management of plastic waste around the world; Vanapalli et al. [27] explore the challenges and strategies in their effective management; in the field of household waste, Ikiz et al. [3] examine the impact of COVID-19 on the treatment of this type of waste in residential buildings; and Ouhsine et al. [25] assess this impact on the generation of this waste and on consumption habits.

Shove [28] points out that both consumption and domestic practice establish a close relationship in the reproductive circuit of what the subjects consider within normality and, in the same way, common ways of life. Thus, consumer behavior in a pandemic scenario is linked to different issues and an understanding of how these issues can affect this behavior also requires a consideration of the social nature, thus proposing the lens of practices for the study of practices for reusing plastic supermarket bags.

*2.2. Practice Theories*

Practice theories is a school of thought [29] in which 'practice' represents a fundamental aspect for social life [30,31], and its sets of constituent actions have their understanding based on sayings and actions related to these actions [32]. The 'practice' refers not only to organized actions, it also inserts these sequenced human activities in social and material contexts [29]. Furthermore, practices have gradually been conceiving their reproduction spaces in social life [30].

Because they are applied and studied in different areas, practice theories present different approaches, standing out as one of the few consensuses where the character of diversity concerning the approach to practices is found [31]. From the last decade of the last century to the present time, its theoretical framework has been developed in numerous areas such as education, geography, history, art, sociology, political science, and organizational studies [29], with works by authors such as Giddens [33], Bourdieu [34], Schatzki [35], Reckwitz [36], Nicolini [37,38], Gherardi [39] e Shove, Pantzar, and Watson [40].

Multiple are also the phenomena to which studies of this approach are concerned, such as consumption, learning, teaching, professions, migration, organizations, international relations, sustainability, and energy use [29]. Warde et al. [41] highlight the importance of the role that this theory has played in the evolution of discussions in the area of consumption.

In consumption, the analysis of practices is often operationalized based on its constituent elements. There are different ways of structuring these elements, and in this paper, we utilize Shove et al. [40]. The authors explain that the elements of practices are formed by three sets: the materials (i.e., the "materiality" that allows the practice, objects, infrastructure, tools, the physical part of the equipment and the body itself), meaning (i.e., mental activities, emotions, and motivational knowledge, teleactivity), and competence

(i.e., socially shared understandings of good and adequate performance, and the skills necessary for that performance).

Since the solution to the problem of plastic production and use requires the involvement of individuals and covers the use of materials, infrastructure, the psychological effects of interventions, efficiency strategies [16], as well as knowledge, technology, and behavior change [42], different elements of the practices are mobilized in their use. Hagberg [43] uses the elements of the practice to investigate the trajectory of shopping bags, be it paper or plastic, and their contributory role in the formation of a bundle of practices, as well as their transformations resulting from the impacts of these bundle of practices of consuming bags. Considering the timeliness and emergence of studies regarding the COVID-19 pandemic, several gaps for investigation are identified, such as changes in consumer practices, the acquisition of new habits or the formation of new consumption patterns, as well studies of social practice in Global South countries.

## 3. Materials and Methods

This study is part of a research that addresses emerging practices in the combat of COVID-19. We use an ethnographic-inspired approach [44] to achieve its objective of analyzing how the COVID-19 pandemic has affected the practices of reusing plastic grocery bags through the lens of practice theories. Amid the educational principles of reducing the amount of waste (i.e., policy of the 5 Rs: Rethink, Refuse, Reduce, Reuse and Recycle [22]), this ethnographic-inspired study will focus on the performances related to the procedures applied for the secondary reuse of plastic supermarket bags, that is, their reuse.

The ethnographic approach has been used in investigative studies of anthropology since the beginning of the 19th century and, since the middle of the 20th century, it has gained ground in the social sciences [45]. This approach seeks to illustrate "a cultural or social life, world or experience" [46].

In this research, data collection took place through records in 'requested diaries' digitally combined with interviews applied during and after the closure of the diaries, in addition to the notes of autoethnography field diaries. Interviewing the informant daily allows for the building of a relationship and possible deepening of the diary [47], in addition to generating, through the combination of the two, an approximation to the participant observation method [48] and the possibility of disclosure of important differences [44].

Between April and July 2020, we applied the 'solicited diaries'. They were guided by the approach of Zimmerman and Wieder [48], where we directed the informants on how the diaries should be executed through a conversation. Subsequently, they made the records in 15 days. At the end of that period, we analyzed the diaries. Then, we carried out a final interview based on the collected data. The use of 'solicited diaries' in the social sciences for a better understanding of personal experiences in everyday life has become increasingly widespread [49].

The diary's topics related to the plastic bag reuse practices in supermarkets sought to investigate how the COVID-19 pandemic has affected these practices. We focused on details of this practice through the initial registration of the autoethnographic diaries field started in March 2020. We made notes about how we experienced pandemic context and we made observations regarding the practices of reusing supermarket bags adopted in the fight against the coronavirus.

With the support of these data, we carried out a pre-test with a group made up of four individuals. From that informant group, we delved into how diary topics could be addressed and how the 'solicited diaries' should be applied. After adjustments, we initiated the 'solicited diaries' with 48 participants after their agreement with the informed consent form and participation in the research.

We invited the informants to report and record images of the procedures adopted after the COVID-19 breakthrough, when supermarket purchases arrived at the residence, whether or not they had established a protocol for cleaning plastic bags in their daily lives,

and what changes occurred concerning the period before the pandemic. They documented their reflections, feelings, knowledge, and artifacts used in the execution of their practices.

The group of research participants was composed of 45 Brazilians living in five Brazilian states (Ceará, Minas Gerais, Pernambuco, Piauí, and the Rio Grande do Sul) captured using the "snowball" technique [50]. After the initial procedures and notes we focus on "saturation", or the point of redundancy of data, presented by the participants (a decisive point in qualitative/ethnographic research, see [51,52]). This saturation is also aligned with [47], the expert explains that the research using diaries depends on the time length of the diary, and a number of ca. 25 diaries fit with our research design. In this way, 30 fit the target profile of this study and participated in the research until the final stage of the proposed period. The age of the informants is well distributed around 30–33 years old, which fits with the Brazilian median age in 2020, i.e., 33.5 [53]. The informants were coded with the letter "I" and numbered from one to 30, which corresponds to the number of participants until the closing interview phase, to preserve their anonymity and confidentiality. Their demographic profiles, as well as the number of people living with them and interviewed during the pandemic period, can be seen in Table 1:

**Table 1.** Demographic profile of interviewees.

| Code | Age | Sex | Country | Number of Residents |
| --- | --- | --- | --- | --- |
| I01 | 20 | Female | Fortaleza/Ceará/Brazil | 2 |
| I02 | 21 | Female | Fortaleza/Ceará/Brazil | 3 |
| I03 | 21 | Female | Fortaleza/Ceará/Brazil | 3 |
| I04 | 24 | Male | Quixeramobim/Ceará/Brazil | 2 |
| I05 | 25 | Female | Fortaleza/Ceará/Brazil | 2 |
| I06 | 25 | Female | Belo Horizonte/Minas Gerais/Brazil | 4 |
| I07 | 26 | Female | Fortaleza/Ceará/Brazil | 3 |
| I08 | 27 | Female | Fortaleza/Ceará/Brazil | 4 |
| I09 | 27 | Male | Fortaleza/Ceará/Brazil | 1 |
| I10 | 27 | Female | Recife/Pernambuco/Brazil | 1 |
| I11 | 28 | Male | Fortaleza/Ceará/Brazil | 1 |
| I12 | 29 | Female | Fortaleza/Ceará/Brazil | 1 |
| I13 | 30 | Female | Fortaleza/Ceará/Brazil | 4 |
| I14 | 30 | Male | Viçosa/Minas Gerais/Brazil | 1 |
| I15 | 32 | Female | Fortaleza/Ceará/Brazil | 0 |
| I16 | 33 | Female | Fortaleza/Ceará/Brazil | 0 |
| I17 | 33 | Female | Fortaleza/Ceará/Brazil | 3 |
| I18 | 33 | Male | Fortaleza/Ceará/Brazil | 4 |
| I19 | 35 | Male | Fortaleza/Ceará/Brazil | 3 |
| I20 | 35 | Male | Fortaleza/Ceará/Brazil | 3 |
| I21 | 35 | Female | Teresina/Piauí/Brazil | 5 |
| I22 | 35 | Female | Porto Alegre/Rio Grande do Sul/Brazil | 0 |
| I23 | 35 | Male | Vespasiano/Minas Gerais/Brazil | 3 |
| I24 | 36 | Female | Fortaleza/Ceará/Brazil | 3 |
| I25 | 37 | Female | Fortaleza/Ceará/Brazil | 2 |
| I26 | 37 | Female | Fortaleza/Ceará/Brazil | 2 |
| I27 | 37 | Male | Fortaleza/Ceará/Brazil | 0 |
| I28 | 41 | Female | Fortaleza/Ceará/Brazil | 1 |
| I29 | 48 | Female | Fortaleza/Ceará/Brazil | 2 |
| I30 | 62 | Female | Santa Maria/Rio Grande do Sul/Brazil | 1 |

In the face of an unusual phenomenon like a pandemic, collecting data is an activity that becomes much more challenging. The first challenge we faced was the observation in loco of the participants in the practices of this study since the present time required the social distance and reclusion of each one in their homes. The second one was the interviewees' perseverance in preparing the diaries. During the data collection, some informants reported symptoms that made them on the list of suspects or were even confirmed with COVID-19, which prevented the continuation of some of them in the research.

In this way, we sought to apply several data collection strategies to overcome these challenges and, in the same way, to correspond to the methodological demands demanded by the practice theory, to observe what happens when the practices are performed [18]. In addition to the written and recorded audio reports sent by the participants through WhatsApp and Telegram, we also used photos and videos, which allowed the observation of the corporal performance of the practices, as well as the artifacts used. In addition to these techniques, we jointly used video calls made via Google Meet. Agafonoff [46] points out that, according to technological evolution, new techniques such as audio and photography recordings have been used in the capture and communication of ethnographic discoveries.

The triangulation of these different data sources, which Yin [54] proposes as a means of validating the same fact or phenomenon, made it possible to investigate the skills, knowledge, and emotions intrinsic to the "doings" and "sayings" of the plastic bag reuse practices supermarkets in the home environment, as well as how these practices were impacted in the context of the COVID-19 pandemic.

The process of analysis of the collected data took place through the content analysis proposed by Bardin [55] of categorical analysis. This process comprises two stages: (i) inventory; (ii) classification. The first one occurred in this research through the isolation of passages, words, and images that pointed to the formation of and changes in the reusing supermarket plastic packaging practices in the period studied. The classification step relied on the organization of data that were isolated into categories according to the elements of competence, material, and meaning that constitute the practices [40] regarding the reuse of plastic supermarket bags in the context of the COVID-19 pandemic. The mentioned steps enabled the study to become operational to meet the proposed objective.

## 4. Results and Discussion

The practices of reusing plastic bags from supermarket purchases have faced new barriers in the COVID-19 pandemic context. From the reports and records obtained from the Brazilian informants, there was a greater tendency to discard this type of packaging and different performances to maintain this practice. To map some ways in which these practices are behaving in the context of isolation, we observed different aspects that referred to the elements of the practices proposed by Shove et al. [40].

At first, the informants reported a greater criterion regarding the space of the home in which initially the plastic bags with the purchases must be left (Figure 1). What in the past did not receive due care, in the current context is a target of attention due to the risk of contamination by the SARS-CoV-2 virus. We can see in the records from the requested diaries that sites close to the entrance and exit door of the residence are the most targeted, either on the floor, the preferred place by most interviewees, or benches and tables arranged as close as possible to the access door. These spatial redefinitions for carrying out the practices under study are associated with the arguments of Schatzki [30] regarding the spatial component of the practice, where the author mentions that places are linked to material entities to make human actions feasible and, in this way, integrate other elements for the execution of the practices.

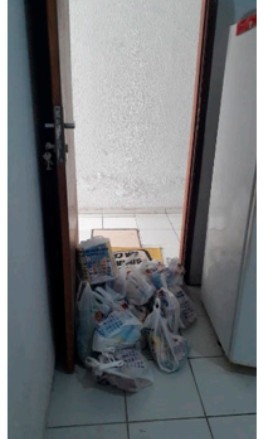 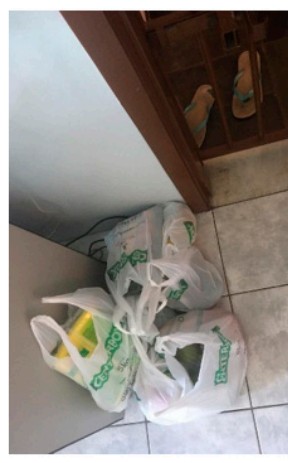 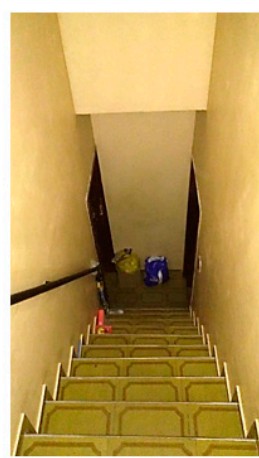 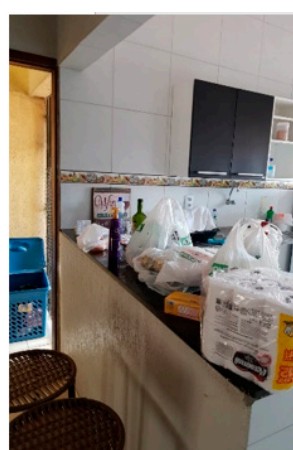

**Figure 1.** Photos of spaces in which products placed in plastic supermarket bags are left. Source: Research data.

Some informants revealed that they started to give preference to spaces outside the house, such as a garden or yard, to prevent the bags from entering the internal environment of the residence, as shown below:

*And then, when I get home, [...] the products I bought [...] we put them out here in the area [...] we remove all the bags and then after we clean the products, we clean the bags. (I07, Ceará, woman, 26 years old.)*

*When things arrive, right, they stay here at the door, because the door is already open to the kitchen, right? So things are coming, my sister does the cleaning at the door, right? She already removes the secondary part [grocery bags, boxes] at the door (I19, Ceará, male, 35 years old.)*

After removing the products purchased from the bags, they were often discarded shortly thereafter (Figure 2):

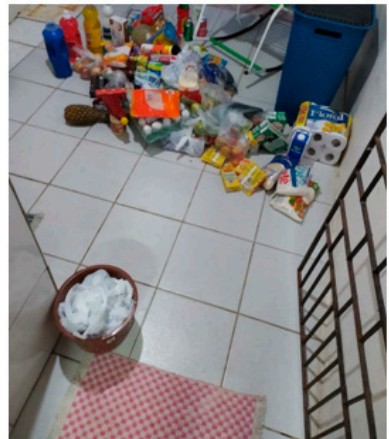 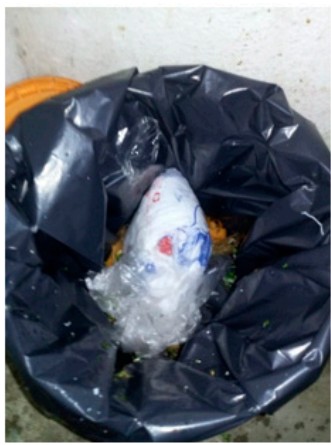 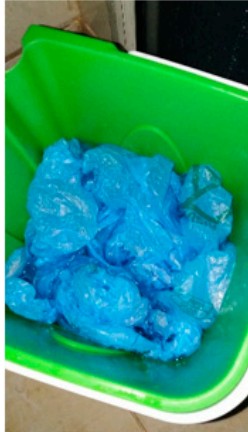

**Figure 2.** Photos of the material elements of the disposable plastic grocery bags. Source: Research data.

That was an activity that did not predominate in the interviewees' homes, but which was carried out promptly, as shown by the following reports:

*And we decided to adopt the system of not going in with the bags or the kitchen anymore, stay in the area, we put it in, take things out of the trunk, right, from the car [...] and put it in another box, in a box of ours right here and she who enters the kitchen, the bags do not enter, they stay outside and [go] straight to the trash so that there is no danger of contamination. (I23, Minas Gerais, male, 30 years old.)*

*When shopping arrived, we were very careful, right? To clean everything, first, remove all plastic bags, then throw everything in the trash. (I02, Ceará, woman, 21 years old.)*

*I just throw the bags away. (I28, Ceará, woman, 41 years old.)*

This increase in the disposal of plastic waste is one of the many harmful effects resulting from the COVID-19 pandemic [2], where the need to promptly deal with the virus can leave sustainability in the background [2,26]. Hence, the lack of development and continuous adjustments of contingency plans to guide the future of plastic, since, according to Vanapalli et al. [27], this loosening in the use of disposable plastic, although momentary, may have the consequence of a permanent change in consumer behavior.

The decontamination process for plastic supermarket bags has a variety of skills and abilities (Figure 3). Even if the bags are not reused, they initially have alcohol or bleach diluted in water sprayed throughout their external extension before removing the products contained in them, as reported by the following informant: *"As soon as you enter the house, you have the table with alcohol, and then I spray 70 alcohol on the bags [...] this bag from outside we throw it in the trash." (I03, Ceará, woman, 21 years old.)*

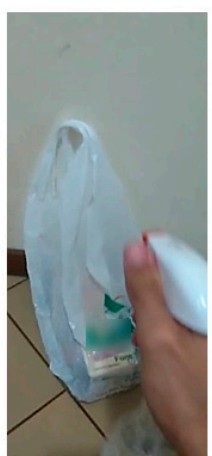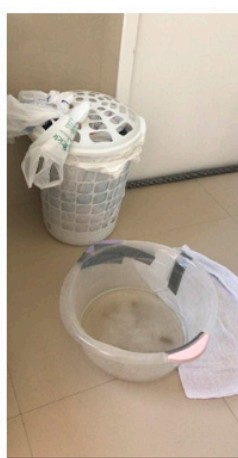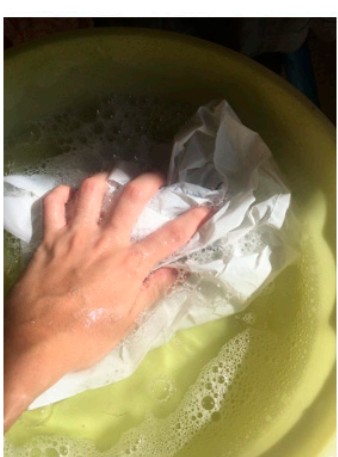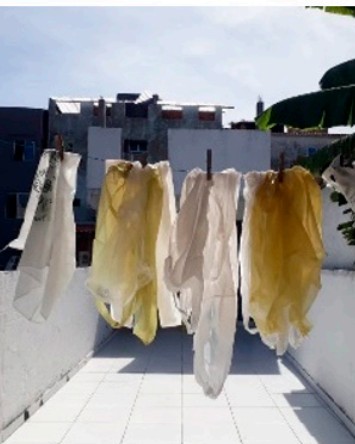

**Figure 3.** Photos of the different procedures adopted in the handling of plastic supermarket bags. Source: Research data.

The next step follows one of the three options: (i) disposal; (ii) storage in a place for later reuse; (iii) or one more cleaning process for reuse. This second cleaning process includes activities such as washing with soap and water or soaking in a bleach solution diluted in water (Figure 3). The different performances reported in the process of cleaning plastic supermarket bags for later reuse are illustrated below:

*All bags are also dipped in the cool box [with a bleach solution diluted with water] and then extended to dry. We use them, generally, to put the garbage. (I11, Ceará, male, 28 years old.)*

*The supermarket bag, I use it to throw the trash out, right? Yeah, so I always use it for trash, but when they arrive with me from shopping, I clean them with alcohol, extend them a little, to get a little wind and dry, then I put it in a bag, and then I use it for the trash, the bathroom, and the kitchen, right? (I22, Rio Grande do Sul, woman, 35 years old.)*

*When we bring everything, we wash the [plastic] bags, soak them in the soap and wash the packaging, each one. [...] I let it soak, it is::: in a bucket with soap and water, we soak it there and then put it to dry on the clothesline because we need these bags to put the trash out. (I05, Ceará, woman, 25 years old.)*

*The bags, we even have them reused for garbage, the supermarket bags, we::/we pass the bleach on everything that arrives, while it is still closed we pass bleach on the outside of the packaging, either box like a bag. The bags, we after the bleach, we leave them, like, on a clothesline and they stay there for two, three days, then we take them to use for the garbage bag, these things. (I18, Ceará, male, 33 years old.)*

*The bags, it is a little complicated at home because the bags, my mother wants to use to throw garbage, you know, throw the garbage away, then what do I do [...] I take it, wash the bag, [...] we wash the bag, let it dry, leave it outside, which is for when you want to put the garbage. (I21, Piauí, woman, 35 years old.)*

*I usually put the packages in a bag and put it in the service area, in that part on the service area balcony. So, they spend a week there and yes, before closing the bag with the packages / with the other bags inside, I spray a lot of alcohol, do it like a stick, and tie it up and leave it there. [...] Then I will get a bag from there that the supermarket was made about ten, fifteen days ago, then here I will use it now. (I29, Ceará, woman, 48 years old.)*

The results demonstrated how the reuse practices are compromised due to the concern of the possibility of transmission of the SARS-CoV-2 virus. These experiences can be examples of what Ikiz et al. [3] highlight as the negative implications for household waste streams, as regards the approach of Vanapalli et al. [27] on the complications of plastic waste management in facing the pandemic.

Mental activities and feelings related to washing the bags for later reuse were manifested in the concern of the possible impacts of the disposal of this product, in addition to negative emotions for having this activity be fulfilled:

*It was on... on TV, I remember it like that, they weren't, they weren't experts, but they made recommendations and then [they said] 'oh, discard the plastic bags' and I think this is very wrong, you know? Because it's plastic, right? And when you buy a large volume, a lot of bags come and then we also chose to clean the bags and put them to dry, so that we can then reuse these bags. (I07, Ceará, woman, 26 years old.)*

*Then I feel frustrated and upset because all life has to spend part of our time, of our day to be able, that is, to clean fruit and bags [...] (I26, Ceará, woman, 37 years old.)*

Other words/expressions that showed the informants' motivation for this practice to be performed were: "avoid polluting the environment", "feel safer", and "protection".

When analyzing the impact of the COVID-19 pandemic on supermarket bag reuse practices through the lens of the practices employed by Shove et al. [40], we identified a sudden increase in its environmental footprint, either due to the increase in its disposal and the consequent increase in the generation of waste or by the insertion of new processes that enable its reuse, whose objective is hygiene in combatting contamination by the virus. These new procedures involve an increase in the consumption of cleaning products, such as bleach and soap, the addition of alcohol in this disinfection routine, and an increase in water consumption, which signals major impacts on the environment through the use of a natural resource that is in danger of scarcity and the release of polluting substances.

Given the repercussions identified, this study highlights the materialities involved in attempts to reuse plastic bags from supermarkets, as well as their performances and feelings, igniting an alert to the possibility of incorporating disposal practices by individuals who said that before the pandemic they had the habit of reusing bags. Because of this, it is evident that there is a need for regulatory incentives, legislation, investments in physical infrastructure, and educational campaigns that disseminate knowledge and awareness regarding the practices related to the correct management of domestic waste in facing the pandemic of COVID-19 and already with a view to the post-pandemic period.

## 5. Conclusions

This study examined the impact of the COVID-19 pandemic on the reuse of plastic bags from supermarkets among Brazilian consumers through the lens of practices. Through the methodological procedure of ethnographic inspiration, it was possible to build, from the daily reports of 30 informants, changes in the treatment given to plastic supermarket packaging in the fight against COVID-19.

These alterations suffered in the performances invariably culminate in environmental impacts, either in the disposal of the plastic bags or in their hygiene for later reuse. For this

observation, the material elements involved in the execution of these practices, the feelings that drive these behaviors, as well as the competence of their practitioners were explored.

We identified that the reuse practices of plastic supermarket bags in the context of the COVID-19 pandemic requires a greater load of material artifacts that lead to an increase in its environmental footprint and, thus, demonstrates different ways in which this context has impacted on production of household waste, the use of water resources, contamination of the environment, in addition to changing to less sustainable behaviors to the detriment of preserving the health of individuals.

These results alert to the challenges that governments, institutions, and individuals will have to face in an attempt to reverse the damaging effects of the pandemic on sustainability goals. Also, it contributes to the literature of consumer behavior in a crisis context, as well as concerning household waste. The results also bring information to the supermarkets and grocery stores about the new role and importance of the plastics bags and the demand for specific cleaning products, so the market could use more robust plastic bags as a differentiation strategy.

As limitations of the study, we could point out that it has been carried out in only a few Brazilian states, and maybe due to the "complex" diaries methodology we did not have many people older than forty years old take part. However, the possibility of future research in face of this factor is its expansion to the states in which it was not carried out. Other suggested study itineraries may be a comparison of the performance of the practices studied after the end of the pandemic, or even among individuals from other countries.

**Author Contributions:** Conceptualization, A.R.G. and J.C.L.; methodology, A.R.G. and J.C.L., formal analysis, A.R.G., A.L. and J.C.L.; investigation, A.R.G.; resources, A.R.G. writing—original draft preparation, A.R.G.; writing—review and editing, A.R.G., A.L. and J.C.L.; supervision, J.C.L.; project administration, A.R.G. and J.C.L.; funding acquisition, A.R.G. and J.C.L. All authors have read and agreed to the published version of the manuscript.

**Funding:** This research was funded by FUNCAP-Fundação Cearense de Apoio ao Desenvolvimento Científico e Tecnológico.

**Institutional Review Board Statement:** Not applicable.

**Informed Consent Statement:** Informed consent was obtained from all subjects involved in the study.

**Data Availability Statement:** The data are not publicly available due to privacy or ethical restrictions.

**Acknowledgments:** The authors would like to thank all the interviewees who participated in this research.

**Conflicts of Interest:** The authors declare no conflict of interest.

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
