# Peer review of "Should I Reuse It or Throw It Out? Analysis of the Management of Household Plastic Waste by Brazilian Consumers during the COVID-19 Pandemic through Practice Lens"

_sustainability, doi:10.3390/su14148512_

Round 1
Reviewer 1 Report
Dear authors,
You have a good topic, but there are some issues that need to be clarified.
lines 30-38 are template
lines 175-178 which are the gaps identified? it is not clear enough.
"Of this universe, 30 fit the target profile of this 223 study and participated in the research until the final stage of the proposed period. "- I understand that the pandemic was a difficult period for collecting data, but still, 30 participants is not leading to relevant results....
Please mention the managerial implications of your study.
Author Response
Dear reviewer,
Thank you very much for the comments.
hereby we hope we answer your questions
- You have a good topic, but there are some issues that need to be clarified. lines 30-38 are template
- Thank you. We have supressed the content for lines 30-38 in the section Introduction.
- lines 175-178 which are the gaps identified? it is not clear enough.
- Thank you for this observation. We added to the section Theoretical background (lines 178-179): “(…) such as changes in consumer practices, the acquisition of new habits or the formation of new consumption patterns, as well as studies of social practice in Global South countries..”
- "Of this universe, 30 fit the target profile of this 223 study and participated in the research until the final stage of the proposed period. "- I understand that the pandemic was a difficult period for collecting data, but still, 30 participants is not leading to relevant results...
- We would like to reinforce that it is qualitative research, analysing “social practices” using an ethnographic inspired approach. Studies in this area used to use the saturation of categories or emergence of regularities as criteria to “stop collecting and processing” data (Lincoln & Guba,1985:352; Patton,2015:454).
- We also empathize that
- So the corpus wasn’t 30 interviews, but material from 30 participants, using solicited diaries, a very specific methodology.
- We understand that in qualitative research the corpus depends on the saturation of information. Around the 30th interviewee, we understand that there was enough material converging for a common practice.
- Nevertheless, some qualitative authors like Morse (2000) consider that, in qualitative studies, in the case of interviews carried out many times with people over time, between 6 and 10 is a sufficient “sample”. We used this author as a parameter.
- Please mention the managerial implications of your study.
- The main managerial implication is to bring a new possible approach to waste management policymakers, as Spaargaren (2011) comments, but also The results also bring information to the supermarkets and groceries about the new role and importance of the plastics bags and the demand for specific cleaning products, so the market could use a more robust plastic bags as a differentiation strategy.

Reviewer 2 Report
Line 30 - 38 on page 1 need to be deleted. The author's forgot to delete the submission instructions.
I will give the authors credit for originality, it is a clever concept. However, I am not sure there is a lot of interest in this topic.
I do not believe the sample size is large enough to have any meaningful insights into how COVID impacted household plastic waste.
There is little to no data presented to support the qualitative data.
Author Response
Dear reviewer,
Thank you very much for the comments.
hereby we hope we answer your questions.
Line 30 - 38 on page 1 need to be deleted. The author's forgot to delete the submission instructions.
- Thank you. We have supressed the content for lines 30-38 in the section Introduction.
I will give the author credit for originality, it is a clever concept. However, I am not sure there is a lot of interest in this topic.
I do not believe the sample size is large enough to have any meaningful insights into how COVID impacted household plastic waste. There is little to no data presented to support the qualitative data.
- We would like to reinforce that it is qualitative research, analysing “social practices” using an ethnographic inspired approach. Studies in this area used to use the saturation of categories or emergence of regularities as criteria to “stop collecting and processing” data (Lincoln & Guba,1985:352; Patton,2015:454).
- We also empathize that
- So the corpus wasn’t 30 interviews, but material from 30 participants, using solicited diaries, a very specific methodology.
- We understand that in qualitative research the corpus depends on the saturation of information. Around the 30th interviewee ,we understand that there was enough material converging for a common practice.
- Nevertheless, some qualitative authors like Morse (2000) consider that, in qualitative studies, in the case of interviews carried out many times with people over time, between 6 and 10 is a sufficient “sample”. We used this author as a parameter.
Alaszewski, A. Using Diaries for Social Research; Sage, 2006
Lincoln, Y. S., & Guba, E. G. Naturalistic Inquiry. SAGE, 1985
Morse, J. M. (2000). Determining Sample Size. Qualitative Health Research, 10(1), 3–5. https://doi.org/10.1177/104973200129118183
Patton, M. Q. Qualitative research & evaluation methods : integrating theory and practice (4th ed.). SAGE, 2015.

Reviewer 3 Report
The Authors sought to determine how consumers behaved in the face of the threat of isolation methods and the social distance in the face of COVID-19 on the consumption of single-use plastic bags. Using the methodological procedure of ethnographic inspiration, they built a picture of changes in the handling of used plastic packaging during the pandemic. Part of the society disinfects these packages in order to use them at a later date. However, too much used plastic bags as disposable packaging pollute the environment. They defined the material elements describing these behaviors, the feelings driving these behaviors and the competences of the respondents. They indicated that this increased the environmental footprint by increasing the amount of municipal waste generated.
How to understand Practice Lenses? In title. Bad translation in my opinion.
Lines 30-38 is probably redundant text.
Line 87 - Therefore, its aim is to study the impact of the COVID-19 pandemic on the reuse of supermarket plastic bags among Brazilian consumers through the prism of practice. What practices are we talking about here? Maybe "the practice of handling used bags"
Lines 205-209: How to understand the registration of ethnographic diaries, which started in March 2019? What was it done for? As you know, the pandemic began in November 2019. This problem should be clarified.
Line 230 – intervieweds , must be interviewers
Line 508: Funding: “This research was funded by FUNCAP” and “The APC was funded by FUNCAP”. Check carefully that the details given are accurate and use the standard spelling of funding agency names at https://search.crossref.org/funding. Any errors may affect your future funding. What's going on here? What kind of financing and who is to make the adjustment? - Reading text?
Author Response
Dear reviewer,
Thank you very much for the comments.
hereby we hope we answer your questions.
- How to understand Practice Lenses? In title. Bad translation in my opinion.
- Seminal authors from the field Theories of Practice of use the term “practice lens”. (Feldman & Orlikowski, 2011; Gherardi, 2009; Hui et al., 2017). Also in the Theories of Practice’s field some authors use “practice lenses” in plural due to the discussion of plural approaches to the social fact using the Theories of Practice (Schatzki) (Moura & Bispo, 2021; Silva & Figueiredo, 2020)
- We changed the Title, also aligned to the demand of other reviewer. And will use in singular, Practice Lens.
- Lines 30-38 is probably redundant text.
- Thank you. We have suppressed the content for lines 30-38 in the section Introduction.
- Line 87 - Therefore, its aim is to study the impact of the COVID-19 pandemic on the reuse of supermarket plastic bags among Brazilian consumers through the prism of practice. What practices are we talking about here? Maybe "the practice of handling used bags"
- we rewrite this part line 87 - 92, to let it more clear.
- Lines 205-209: How to understand the registration of ethnographic diaries, which started in March 2019? What was it done for? As you know, the pandemic began in November 2019. This problem should be clarified.
- Thank you for this observation. In the section Materials and Methods, lines 205-209, we misspelt this information. Actually, the correct year is 2020
- Line 230 – intervieweds , must be interviewers
- Thank you. We corrected the spelling of the word on the line 230.
- Line 508: Funding: “This research was funded by FUNCAP” and “The APC was funded by FUNCAP”. Check carefully that the details given are accurate and use the standard spelling of funding agency names at https://search.crossref.org/funding. Any errors may affect your future funding. What's going on here? What kind of financing and who is to make the adjustment? - Reading text?
- Thank you. We made the adjustments about funding on the line 508
.
Feldman, M. S., & Orlikowski, W. J. (2011). Theorizing practice and practicing theory. Organization Science, 22(5), 1240–1253. https://doi.org/10.1287/orsc.1100.0612
Gherardi, S. (2009). Introduction: The Critical Power of the `Practice Lens’. Management Learning, 40(2), 115–128. https://doi.org/https://doi.org/10.1177/1350507608101225
Hui, A., Schatzki, T., & Shove, E. (2017). The nexus of practices - Connections, constellations, practitioners (A. Hui, T. Schatzki, & E. Shove (eds.)). Routledge.
Moura, E. O. de, & Bispo, M. de S. (2021). Understanding the Practice of School Management through the Perspective of Sociomateriality. Organizações & Sociedade, 28(96), 135–163. https://doi.org/10.1590/1984-92302021v28n9606en
Silva, M. E., & Figueiredo, M. D. (2020). Practicing sustainability for responsible business in supply chains. Journal of Cleaner Production, 251, 119621. https://doi.org/10.1016/j.jclepro.2019.119621

Author Response
Dear reviewer,
Thank you very much for the comments.
hereby we hope we answer your questions.
- The first paragraph of the introduction is out of place; it looks like a review comment or editorial note.
- Thank you. We have suppressed the content for lines 30-38 in the section Introduction.
- Maybe it is a function of the methods/subject/literature and authors are simply, therefore, sticking to it, but the framing of the study and results can be improved. The “impact” of an event is often thought of as a comparison of the “before” and the “after.” Authors clearly seem to be interested in this kind of comparison in the study of uses/reuses of shopping bags, with COVID-19 as the event. However, the “before” does not quite come across in the paper. Authors place great weight on the photos (figures 1 through 3) but these are all during the event (so, the “after”). Where are the corresponding effects (photos or figures) “before” the event? Without them, the comparison is incomplete and, for the “before,” we are left only with the words of the subjects in the study – all good, but it is an incomplete story in academic terms.
- It is not a before-after common comparison. we rewrite it more explicit in line 91.
- The perception of the users (the “sayings””) is fundamental in this type of research. So we ask the participants to report in their diaries their new practices, after the COVID-19 event (lien 220-224). The handle of the bags emerges in the field research, in the way the respondents perceived it as the main procedures change
- Given the point above, the title of the paper is better framed as “Should I Reuse it of Throw it? Analysis of the management of Household Plastic Waste by Brazilian Consumers during the COVID-19 Pandemic,” or something along these lines.
Thank you for this observation. We made the title changes but kept the point that we are using the Social Practices Theories Lens.
- The data in table 1 suggests the subjects of the study skew young. Is there a reason for this? And what do authors think the implications are for their analysis?
- we add a comment about it in the research limitations, but the age of the informants is well distributed around 30-33 years old, which fits with the Brazilian median age at 2020, i.e. 33,5 according to [55]
[55] United Nations Departament of economic and Socila Affairs. World Population Prospects.. Brazil. https://population.un.org/wpp/Graphs/DemographicProfiles/Pyramid/76 . 2019

Round 2
Reviewer 1 Report
Dear authors
From my point of view it is ok
Reviewer 2 Report
The paper is improved.